

# 1    Earth Observations from the Moon surface: dependence on lunar libration

Nick Gorkavyi[1], Nickolay Krotkov[2], Alexander Marshak[2]
[1] Science Systems and Applications, Inc., Lanham, MD, USA
[2] National Aeronautics and Space Administration (NASA), Goddard Space Flight Center (GSFC),
Greenbelt, MD, USA
**Correspondence**: Nick Gorkavyi (nick.gorkavyi@ssaihq.com)
*Abstract. Observing the Earth from the Moon has important scientific advantages. The angular*
*diameter of the Earth as seen from the Moon surface is $1.9° \pm 0.1°$ (the angular size varies due to*
*the change in the distance between the Earth and the Moon). The libration of the Moon in latitude*
*reaches an amplitude of 6.68° and has a main period of 27.21 days (or 653.1 hours). The libration*
*of the Moon in longitude, reaching 7.9°, has a period of 27.55 days (or 661.3 hours). This causes*
*the center of the Earth move in the Moon's sky in a rectangle measuring $13.4° \times 15.8°$. The*
*trajectory of the Earth's motion in this rectangle changes its shape with a period of 6 years. This*
*apparent librational movement of the Earth in the Moon's sky complicates observations of the*
*Earth. The paper proposes to turn this disadvantage into an advantage and place a multi-slit*
*spectrometer on the Moon surface on a fixed platform. The libration motion and the daily rotation of*
*the Earth will act as a natural replacement for the scanning mechanism.*

## 20    1 Introduction


The scientific benefits of observations from the Moon for the Earth, exoplanet and astrophysics
studies are discussed in several recent papers (Marshak et al., 2020; Gorkavyi et al., 2021; Boyd et
al., 2022). Although current Earth-observing satellites can produce high-resolution images, Low
Earth Orbit (LEO) sensors can only scan a small portion of the globe at a given time, while
Geosynchronous Equatorial Orbit (GEO) sensors can provide temporally continuous, though lower-
resolution observations of a significant, but fixed, portion of the Earth's disk. The Earth
Polychromatic Imaging Camera (EPIC) on the Deep Space Climate Observatory (DSCOVR) clearly
stands apart, observing the entire Sun-illuminated Earth from the L1 Sun-Earth Lagrange point
(Marshak et al., 2018). The L1 location, however, limits phase angles to a nearly backscattering
direction (a phase angle between 2° and 12°). A compact, lightweight, autonomous camera and
spectrometer on the Moon's surface offers a unique opportunity to complement these observations
and image the full range of Earth phases, potentially advancing Earth science in many ways
(Marshak et al., 2020; Gorkavyi et al., 2021):
1. observing ocean/cloud glint reflection for different phase angles;
2. comprehensive whole-globe monitoring of transient volcanic and aerosol clouds, including the
strategically important (for climate studies) polar regions not covered by GEO;
3. detecting of polar mesospheric and stratospheric clouds;
4. estimating the bidirectional surface reflectance factor (BRF) and full phase-angle integrated
albedo;
5. monitoring and quantifying changes in vegetated land;
6. simultaneous imaging of the day and night parts (i.e., the twilight zone) during crescent phases of
the Earth and shadowed parts illuminated by the Moon.





The first telescopic image of the Earth from the Moon was obtained during the expedition
Apollo 16 in 1972 using an ultraviolet telescope (Carruthers and Page, 1972) – see Fig.1. In later
years, prospects for lunar observations of the Earth have been discussed in many papers (e.g., Foing,
1996; Moccia and Renga, 2010). Observations of the Earth with instruments mounted on the Moon
have been actively discussed in recent years (Hamill, 2016). Impressive prospects for observing the
Earth from the Moon in the visible spectrum are demonstrated in the pictures taken by the Lunar
Reconnaissance Orbiter (LRO) in 2015 (Fig. 2).

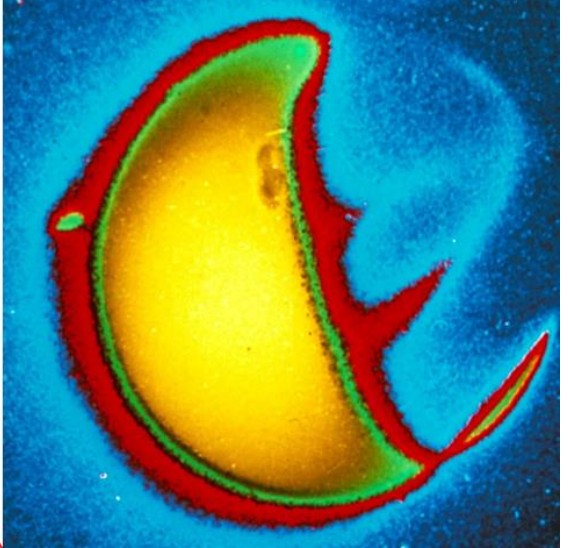


**Figure 1. Left:** A Far-Ultraviolet Camera/Spectrograph was operated on the lunar surface during the
Apollo 16 mission, April 1972 (Credits: NASA/Apollo 16). **Right:** The Earth, photographed in far-
ultraviolet light (1304 angstroms) by astronaut John W. Young. Credits: G. Carruthers (NRL) et
al./Far UV Camera/NASA/Apollo 16; based on the image AS16-123-19657 (Mason, 2019).
One of the objectives of the Chinese space program is Moon-based observation of the Earth
(Li et al., 2019; Guo et al, 2019). A lunar lander Chang'e-3 (landed on the Moon in 2014 and is still
working) is equipped with a 5-cm ultraviolet telescope and extreme UV camera and studied changes
in the Earth's plasmasphere in the UV range (He et al., 2016).
One of the tasks facing the US Artemis program is: "Use the Moon as a platform for Earth-
observing studies… The observations from the Moon will have higher resolution than would similar
observations made from L1. Myriad science investigations targeting topics such as lightning, Earth's
albedo, atmosphere, and exosphere…, the oceans, infrared emission, and radar interferometry may
be accomplished from the surface of the Moon. The Moon also offers a unique vantage point for
full-disk observations…" (Artemis III Science, 2020).
Because of tidal locking, the Moon's rotation around its axis is synchronized with its orbital
rotation around the Earth. Therefore, the Moon always faces the Earth on one side, and the task of
observing the Earth from the Moon seems simple: the Earth must hang motionless in the lunar sky,
rotating around its axis. In reality, the Earth moves along a complex trajectory in the sky of the
Moon due to lunar librations in latitude and longitude.  On one hand, the librations of the Moon
cause the Earth to shift from the field of view of the lunar telescope, which forces one to turn the
telescope to track the Earth observable movement in the Moon sky (guiding); on other hand, for a





fixed slit spectrometer, the librations of the Moon can be useful, because they make the Earth move
through the fixed field of view of the instrument (across the slit). These librations can serve as a
natural mechanism for scanning the Earth when observed from the Moon, which allows the use of
slit spectrometers and telescopes on a fixed platform. This paper takes into account lunar librations
and analyzes the conditions for observing the Earth from the Moon.

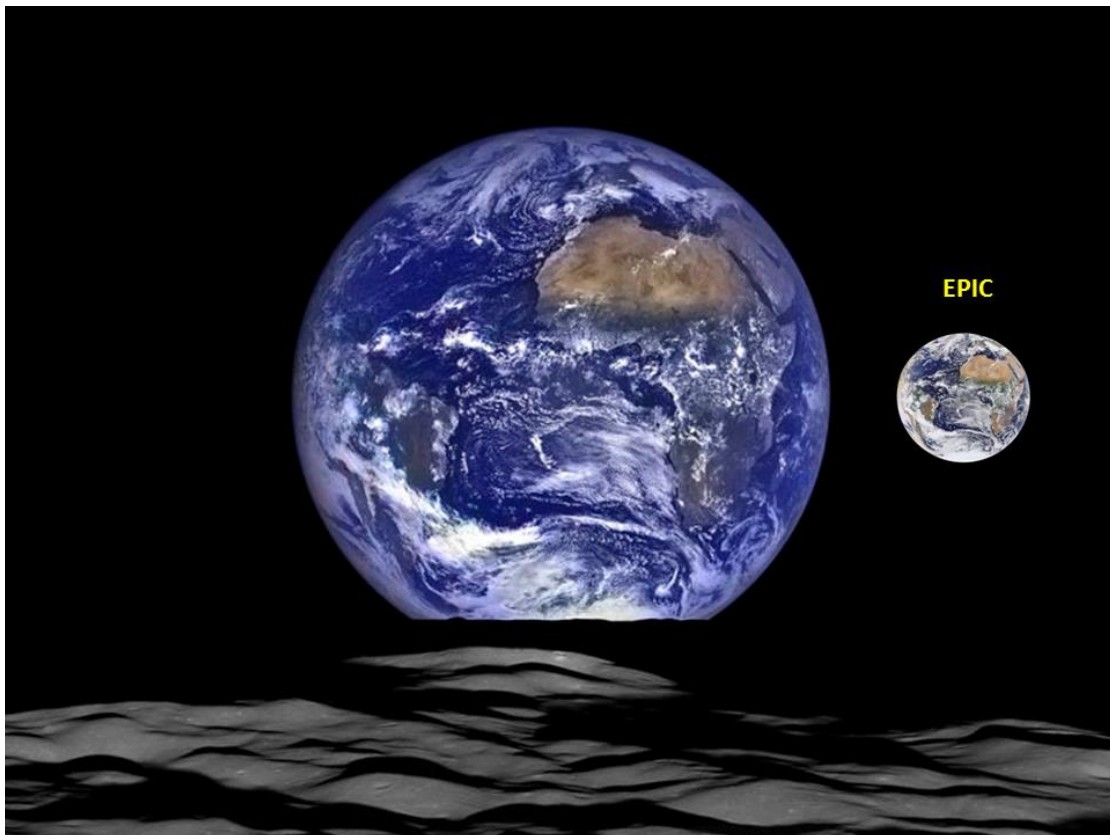

**Figure 2.** A unique view of Earth from the LRO's vantage point in orbit around the Moon (October
12, 2015). LRO was about 134 km above the Moon's farside crater Compton (55°N, 104°E). The
photograph is a combination of images in seven color bands from a wide-angle camera (WAC) and
black and white images from two narrow angle linear pushbroom cameras (NACs) with the linear
(one-dimensional) array from 5064 elements. Each NAC camera has a field of view of 2.86°. Image
Credit: NASA/GSFC/Arizona State University (https://www.nasa.gov/image-feature/goddard/lro-
earthrise-2015). LRO data (Burns et al., 2012; Keller et al., 2016) can be of great help in planning
Earth observations from the lunar surface. Insert shows an image of the same part of the Earth taken
by DSCOVR/EPIC on the same day (October 12, 2015).
**2 Librations of the Moon**





Observations of the Earth from the Moon require accounting for the geometry of the relative
position of the centers of the Earth and the Moon, the inclinations of their axes, and libration effects
(Meeus, 1991,2000; Guo et al., 2018; Xu and Chen, 2019; Huang et al., 2020).
**Libration of the Moon in latitude.** The angle between Moon's axis of rotation (NS) and the normal
to the plane of its orbit around Earth ($PP'$) is 6.68° (Fig. 3). This causes the libration of the Moon in
latitude with the same amplitude and with a period of draconic month $T_D$=27.21222 days or
653.0933 h  (the interval between consecutive passages of the Moon through the same node of the
orbit; an orbital node is either of the two points where a lunar orbit intersects an ecliptic plane to
which it is inclined) – see, for example, Meeus (1991, 2000). As a result of this inclination, parts of
the Moon polar regions are accessible for observations from the Earth.

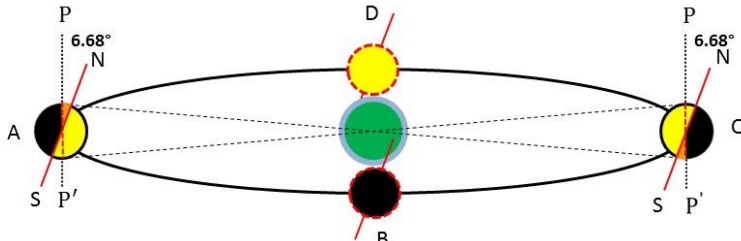

**Figure 3.** Libration in latitude results from an inclination of 6.68° between the Moon's axis of
rotation (NS) and the normal to the plane of its orbit around Earth ($PP'$). The hemisphere of the
Moon that is visible from Earth at point D is marked in yellow; black is invisible hemisphere at
point B. Additional areas of the lunar surface that become available for observation at points A and
C are marked in orange. View from the point close to the ecliptic plane.
**Libration of the Moon in longitude.** The Moon moves around the Earth in an elliptical orbit with
an average eccentricity (or deviation of an orbit from circularity) $e = 0.055$ (it varies between
0.0255÷0.0775) and a period of anomalistic month $T_A$=27.55455 days or 661.3092 h - the interval
between consecutive passages of the Moon through the perigee $r_{min} = a(1 - e)$ or the apogee
$r_{max} = a(1 + e)$ of its orbit, where $a$ is the semi-major axis $a$=384,399 km. (Fig. 4). This causes
libration in longitude with an amplitude of 7.9° (see, for example, Meeus (1991, 2000).
The longitudinal libration consists of two components:
1. An ellipse with a small eccentricity (in the first approximation) can be described as a
circle around the Earth, which is shifted from the center of the circle (point $O$) by the distance
$OE = ea$ (see Fig. 4). This displacement leads to the fact that the observer from the Earth begins to
see part of the lateral surfaces of the Moon (Fig. 4).
2. If the Moon was moving along an orbit with a uniform speed, then its visible part would
always be directed to the center of the orbit (point $O$ in Fig. 4). But the speed of the moon changes
due to the ellipticity of the orbit. If at perigee the Moon is turned to the Earth as in Fig. 4 at point C,
then in a quarter of the anomalistic month $T_A/4$, it should turn 90° counterclockwise at point D (see
the red solid arrow at point D). But due to the high velocity along the orbit segment CD, the Moon
arrives at point D faster than $T_A/4$, so the Moon does not have time to turn 90° (see the red dashed
arrow at point D). This further increases the area of the Moon's surface visible from Earth. On
segment DA, the Moon's orbital speed slows down, and the Moon has time to turn 180° at point A.
With the slow motion of the Moon along segment AB, the Moon has time to rotate around its axis





by more than 90°, which again increases the surface area available for observations from the Earth.
On segment BC, the Moon's orbital velocity increases: the Moon passes through the BC segment
faster than $T_A/4$, so it does not have time to turn 90 degrees and, as a result of this lag, the Moon
returns to its initial position at point C.

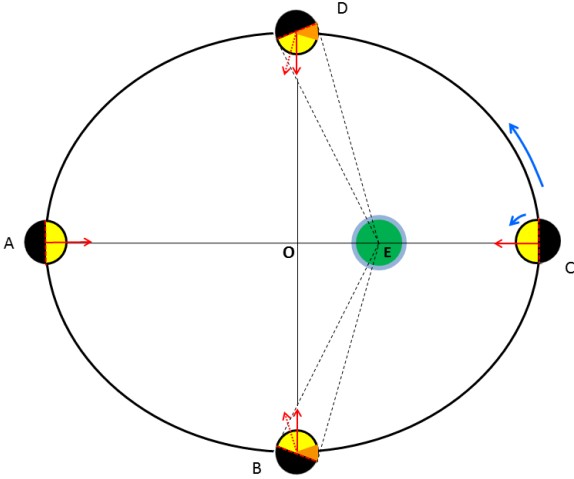

**Figure 4.** Libration in longitude results from the eccentricity of the Moon's orbit. It can reach 7.9° in
amplitude. The point A is the apogee of the lunar orbit ($AE = r_{max}$); C – the perigee ($EC = r_{min}$);
$AO = a$; $OE = ae$; B and D are co-vertices; the semi-minor axis $OB = OD = a\sqrt{1 - e^2}$. Additional
areas of the lunar surface that become available for observation are marked in orange. View from the
North pole.
Figure 5 shows the librations of the Moon in longitude and latitude for the 2022 (Espenak, 2021).
The selenographic coordinate system repeats the Earth's, therefore, the selenographic center of the
Moon's disk is the intersection point of the lunar equator and the lunar prime meridian. The
selenographic zero corresponds to the average position of the center of the visible disk of the Moon.
At any particular moment in time, the center of the visible disk of the Moon can shift from the
selenographic zero due to libration. If the apparent center shifts along the lunar equator, then we call
this shift the relative longitude of the libration; if it shifts along the meridian, then we call this shift
the relative latitude of the libration. In other words, latitude, and longitude libration (or relative
libration) is the visual displacement of the selenographic center of the Moon's disk (0° lunar latitude
and 0° lunar longitude) relative to the center of the visible disk of the Moon. Libration in longitude
correlates with variations in the Earth-Moon distance and changes more strongly with time than
libration in latitude.

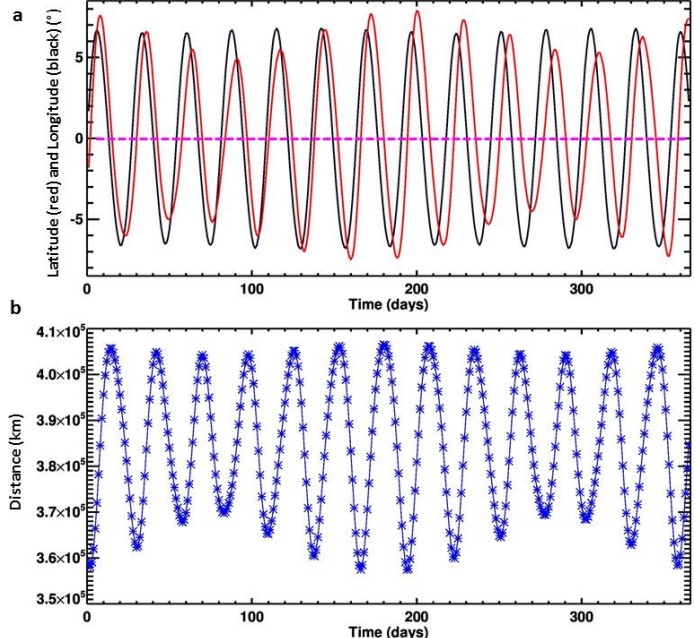

**Figure 5.** Variability of the Moon orientation and orbit during the 2022 starting from January 1, 2022 (Espenak, 2021). The distance between the Earth and the Moon is measured between the centers of the bodies, so it does not depend on libration, which is measured in angles relative to the centers of the bodies.
**(a)** longitude libration (red) and latitude libration (black). The straight dashed line corresponds to the case of zero libration, that is, when the center of the visible lunar disk coincides with the zero point of selenographic longitude and latitude; **(b)** the Earth-Moon distance. The time-averaged distance between the centers of Earth and the Moon is 385,000 km; the minimal distance is 356,500 km and the maximal distance 406,700 km.

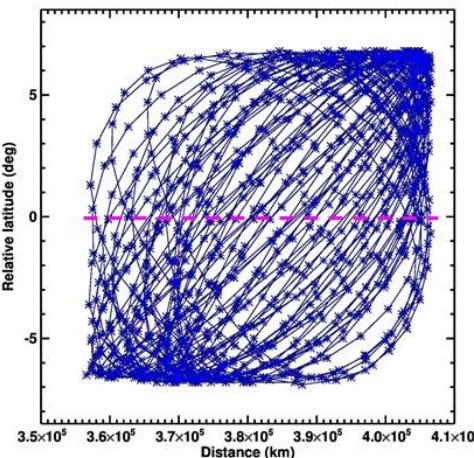
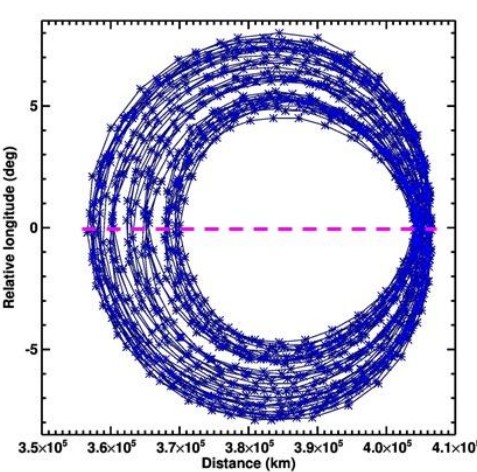

166

**Figure 6.** The relationship between the Earth-Moon distance and lunar librations in latitude **(left)**
and longitude **(right)** during the 2022-2027 starting from January 1, 2022 (Espenak, 2021). The
straight dashed line corresponds to the case of zero libration. The change in the distance between the
Earth and the Moon depends on the eccentricity of the lunar orbit, so it does not depend on the
degree of libration of the Moon.

**3 Visual librations of the Earth**

Obviously, the discussed librations of the Moon are directly related to the observation point on the
Moon surface: if the observation point on the Moon changes its angle relative to the Earth-Moon
line, then the Earth also changes its position in the lunar sky by the same amount, but different sign
(in an approximation where the size of the Moon can be neglected compared to the Moon-Earth
distance). In other words, if the lunar telescope raises its line of sight up by +5 degrees above the
line connecting the centers of the Earth and the Moon, then the Earth goes down from the telescope's
line of sight by -5 degrees.
An observer can see the Earth from any point in the Moon's visible hemisphere (Fig.7). The location
of the observer will affect the i) position of the zero point of Earth libration in latitude and longitude
and ii) orientation of the trajectory of the apparent libration of the Earth in the sky of the Moon.
From the point of view of an Earth observer, lunar librations in latitude and longitude are measured
as a relative displacement from the lunar zero longitude and longitude - that is, from the point of the
lunar disk taken as the zero point and located in the center of the visible disk of the Moon, near the
crater Möstig A. From the point of view of a lunar observer located near this crater at the
intersection of the lunar equator and the lunar zero meridian (see point O in Fig.7), the Earth hangs
over a given point on the lunar surface (at the zenith). Therefore, if the observer moves away from
this point along the lunar meridian, for example, to the North pole (point N in Fig.7), then the
apparent position of the center of the Earth will also shift, moving to the horizon. When the observer
is at the lunar pole, the Earth will hang on the horizon. If the observer goes again to the equator, but
not along the zero meridian, but along the meridian with a longitude of 90° (for example, to West,



see points NW and W in Fig.7), that is, along the border between the visible and invisible
hemispheres of the Moon, then the Earth will remain hanging above the lunar horizon, but will
change the apparent tilt of its axis of rotation. For an observer at the Moon's equator (points W, O, E
in Fig.7), the Earth's axis will tilt 90°, that is, the Earth will "lay on its side."

**Figure 7.** Observers in different points of the Moon's visible hemisphere. Image of the Moon –
LRO (NASA/GSFC/Arizona State University) https://www.nasa.gov/feature/goddard/2020/moon-
more-metallic-than-thought. Observer: photo of John W. Young, commander of the Apollo 16 lunar
landing mission (NASA). It is shown how, from the point of view of different observers on the
Moon, the crescent of the Earth is oriented, indicating the Earth's poles. The yellow squares (with
blue top and violet bottom lines) show the astronaut's vertically oriented field of view (FOV) and the
crescent of the Earth that he sees in this FOV (or in frame of the camera).
When the observer reaches the South Pole (point S in Fig.7), the Earth will be turned 180°
relative to it. Similar changes will occur with the trajectories of the Earth in the sky of the Moon.
Libration of the Moon sets the trajectory of the Earth in the lunar sky described by relative latitude
and longitude (relative to the point of zero libration, marked with a black dot in Figure 8). The shape
of this trajectory (see the red trajectories in Figure 8) is strictly defined and does not depend on the
position of the observer on the Moon's surface. But the height of the point of zero libration above the
horizon depends on the position of the lunar observer, as well as the orientation of the libration
trajectory, that is, the rotation of the visible libration trajectory around this point of zero libration.
An analogy is a picture hanging on the wall of a room. The pattern in the picture does not depend on
the position of the observer, but he can stand on his head and completely change the orientation of
the pattern relative to his field of vision.
We can take the latitude and longitude of the libration of the Moon (Giesen, 2018; Espenak,
2021) (Fig. 5) and plot the positions of the Earth in the sky of the Moon for each day (Fig. 8). Each
dot in the Figures 8 abc represents the latitude and longitude for a particular day,




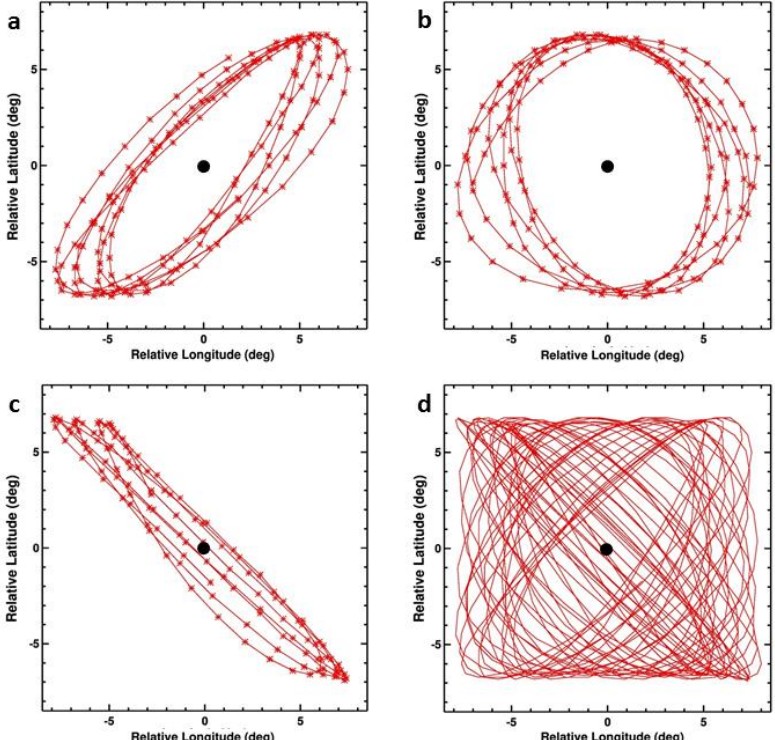

**Figure 8.** Visual librations of Earth in the Moon sky during **(a)** first 6 months 2022; **(b)** first 6
months 2023; **(c)** July-September 2024; **(d)** 2022-2024 (the dots are deleted). The latitude and
longitude of the libration of the Moon (Giesen, 2018; Espenak, 2021) were converted to the relative
Earth libration angles using a sign change. The positions of the Earth in the sky of the Moon are plot
in increments of a day. If there were no lunar libration, then the Earth would be at the black dot in
the center of the figures.

Figure 8a shows the visual position of the Earth for the first half of 2022. Figure 8b shows
the apparent libration of the Earth for the first half of 2023, and Fig. 8c - for 4 months (July-
September) of 2024. Figure 8d shows the trajectory of the Earth in the sky of the Moon for three
years (2022-2024). The orientation of the libration pattern in Fig. 8 corresponds to the position of
the observer on the line of the zero meridian S-O. At point O, the zero libration point is above the
observer's head, and when the observer moves to the South Pole (point S), the zero libration point
shifts to the horizon.

It can be seen that the shapes of the curves along which the Earth moves in the sky of the
Moon change noticeably during 3 years.

The movement of the Moon around the Earth can be characterized by three periods:

- Draconic month $T_D$=27.21222 days or 653.0933 h (the period of movement relative to the
  stars)
- Anomalistic month $T_A$=27.55455 days or 661.3092 h (the period of movement relative to the
  perigee)





• Sidereal month $T_S$=27.32166 days or 655.7198 h (the period of movement relative to the
ascending node)
A beat is an interference pattern between two slightly different frequencies, perceived as a
periodic variation in amplitude whose rate is the difference of the two frequencies. As a results of 3
slightly different lunar periods we have 3 different beats or precession frequencies.
The apsidal precession period is $T_{SA} = 8.85$ years and is found by the formula

$$\frac{1}{T_{SA}} = \frac{1}{T_S} - \frac{1}{T_A} \tag{1}$$

The nodal precession period is $T_{DS} = 18.6$ years and is found by the formula

$$\frac{1}{T_{DS}} = \frac{1}{T_D} - \frac{1}{T_S} \tag{2}$$

The librations of the Moon in latitude and longitude follow to a six-year cycle, when the
major axis of the lunar orbit has performed one complete revolution with respect to the line of nodes
(Meeus, 1991, 2000; Giesen, 2018)

$$\frac{1}{T_{DA}} = \frac{1}{T_D} - \frac{1}{T_A} \tag{3}$$

with a period of $T_{DA} = 2190.34$ days or 6 (more precisely, 5.99667) anomalistic years (365.259636
days each). All three periods of precessions are connected:

$$\frac{1}{6.00} = \frac{1}{18.6} + \frac{1}{8.85} \tag{4}$$

Figure 9a shows the position of the center of the Earth in the lunar sky for six years (2022-2027).
Figure 9b shows the statistics of the distribution of the 2191 positions of the center of the Earth for
this period. The average distribution density of the center of the Earth in squares 1°×1° (or the
number of entries of the center of the Earth into this square for 6 years) $N$ =2191 positions/255
pixels = 8.6; in reality $N$ ranges from 0 to 34.

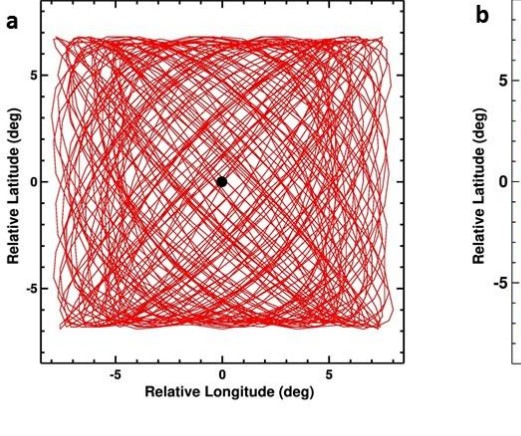
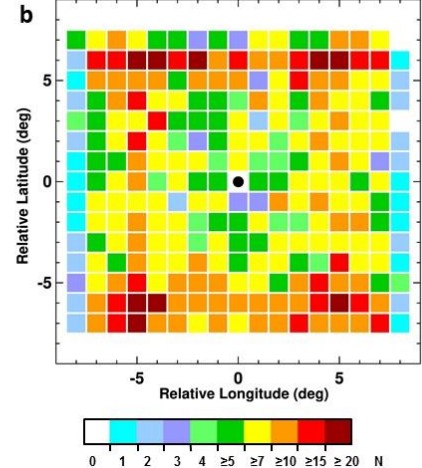


**Figure 9.** Visual libration of Earth during 2022-2027 (6 years, 2191 days or positions): **(a)**
Trajectory of the center of the Earth in the sky of the Moon; **(b)** Statistics of the distribution of the



2191 positions of the center of the Earth for 6 years. *N* - the number of entries of the center of the
Earth into each 1°×1° grid' cell for this period.

## 4 Multi-slit spectrometer on a fixed platform

The angular velocity of a point on the Earth's surface in the field of view of the sensor is caused by
two comparable factors: the rotation of the Earth around its axis and lunar libration, which causes a
shift in the center of the Earth. The rotation of the Earth around its axis is a well-studied process, but
librations of the center of the Earth in the lunar sky are poorly understood and raise many questions.
When observing the Earth through the slit of the spectrometer, it will be necessary to take into
account both the displacement of the center of the Earth and the Earth rotation.
The librational apparent motion of the Earth must be taken into account when observing from the
Moon and can also become a natural substitute for scanning (Figs. 10-11). It is proposed to install
two fixed mount instruments on the Moon surface, directed towards the Earth:
1.  A hyperspectral sensor (UV, Vis, NIR, IR) will observe Earth passing through fixed vertical
slits. The librations of the Moon and the daily rotation of the Earth will serve as a natural
scanning mechanism for this spectrometer. This multi-slit spectrometer can be similar to the
six-slit hyperspectral Limb Profiler (LP) on the OMPS aboard Suomi National Polar
Partnership (S-NPP) LEO satellite, as well as the single-slit hyperspectral Ozone Monitoring
Instrument (OMI) on the NASA Earth Observing System (EOS) Aura satellite. Each LP slit
uses approximately 1/6 of the detector matrix. A multi-slit spectrometer for observing the
Earth from the surface of the Moon can have 6-8 slits, wich field of views are shifted by
2.5°. Since the maximum angular size of the Earth is about two degrees, the angular distance
between the lines of sight of neighboring slits must be greater than the angular diameter of
the Earth, so that light from the Earth does not hit two slits at the same time. Each slit can
use the entire matrix because they scan the Earth at different times in turn and do not
interfere with each other.
2.  A wide field-of-view (WFOV) ~18°-20° camera will continuously image the Earth in any
points of trajectory, including a part of the lunar surface with a true-color calibration target.

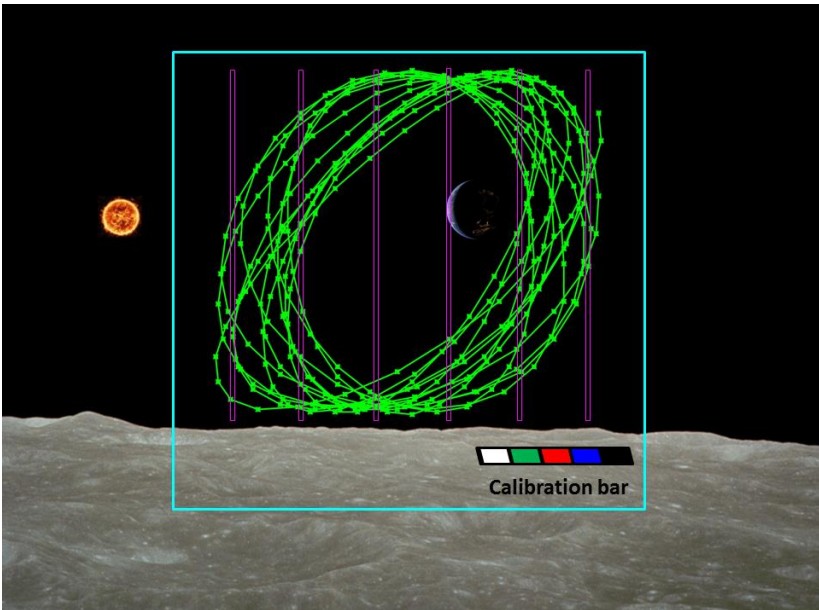

**Figure 10.** Visual positions of the Earth' center during 2026 (green line) from the South Pole of the
Moon (point S in Fig. 7) and possible positions of slits of the spectrometer (violet). Blue square is
~18°-20° FOV of fixed mount camera. The calibration bar is used to calibrate color images from a
wide-angle camera. Lunar surface is from the photo taken by NASA/Apollo.

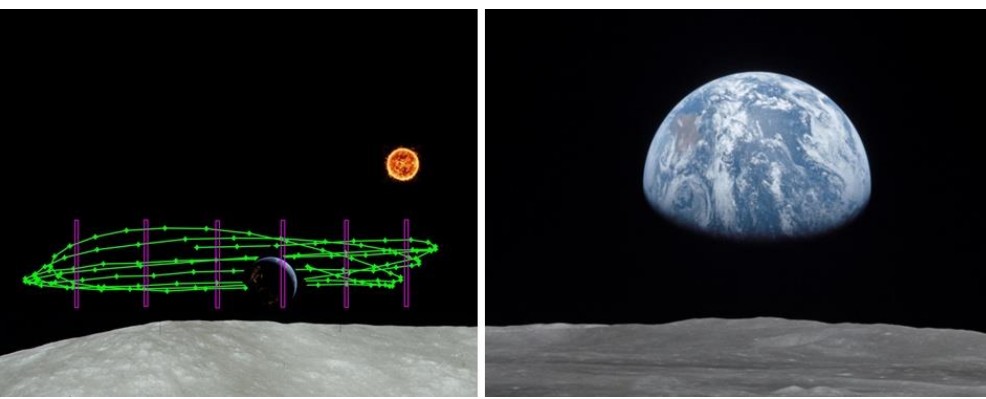

**Figure 11. Left:** Position of the Earth center during July-September 2024 (green line) for an
observer located on the edge of the visible hemisphere of the Moon in the region of middle ~45°N
latitudes (point NW in Fig.7). Violet lines are positions of slits of the spectrometer. **Right:** View of
Moon limb with Earth on the horizon, Mare Smythii region (3°N,85°E), July 20, 1969.
NASA/JSC/Apollo 11, AS11-44-6551
([https://eol.jsc.nasa.gov/SearchPhotos/photo.pl?mission=AS11&roll=44&frame=6551](https://eol.jsc.nasa.gov/SearchPhotos/photo.pl?mission=AS11&roll=44&frame=6551)). Lunar
surface is from the photo taken by NASA/Apollo

It should be noted that the longitude of the observation point on the Moon affects the orientation of
the visible trajectory of the Earth in the sky of the Moon. For example, the diagonally elongated
trajectory of the Earth for July-October 2024 (Fig. 8c, for the case of lunar longitudes near 0°, for



observer in the point S in Fig.7) will have a different orientation when observed from the zone of
lunar longitudes of about 45°N for observer in the point NW in Fig.7 (Fig. 11, left). The orientation
of the Earth from a point located near the equator of the Moon is shown in Fig. 11 (right). The
orientation of the Earth's libration trajectories in the sky of the Moon will change accordingly (see
Fig.12). The sun in the region of the lunar poles moves almost parallel to the horizon, and in the
region of the lunar equator it passes through the zenith, descending vertically to the horizon or rising
from it.

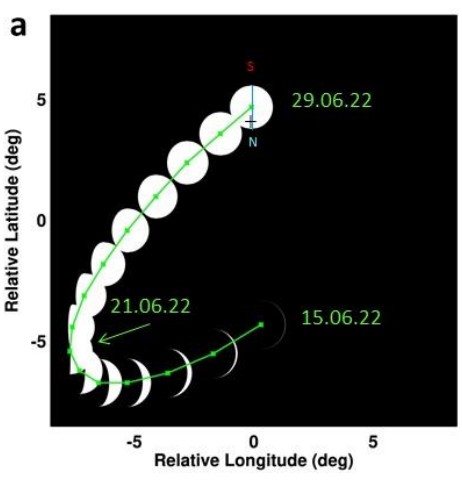

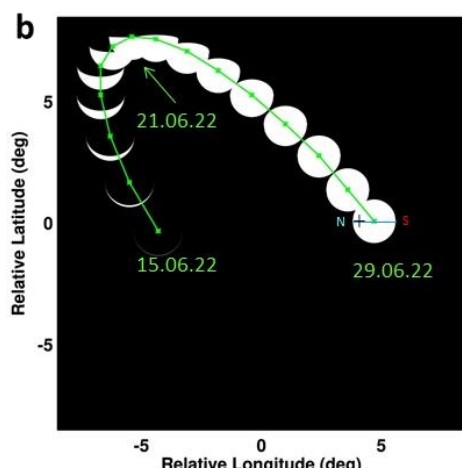


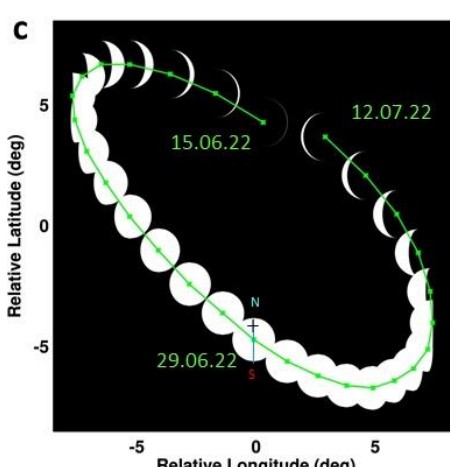

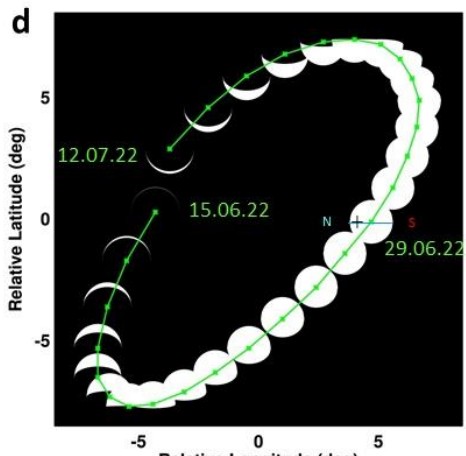

**Figure 12. (a)** Position of the Earth center and Earth' phases during June 15-29, 2022 (15 points,
green line) for an observer located on the lunar South Pole (the point S in Fig.7). The Earth is almost
completely illuminated on June 29, 2022, while its axis is tilted to the Sun at an almost maximum
angle (summer in the Northern Hemisphere), which creates good conditions for observing the
Earth's North Pole (marked with a cross and the letter N).
**(b)** Earth' phases and visible trajectory for same period for an observer on the equator near
longitude 90°W (the point W in Fig.7).
**(c)** Position of the Earth center and Earth' phases during June 15-July 12, 2022 (28 points, green
line) for an observer located on the lunar North Pole (the point N in Fig.7).
**(d)** Earth' phases and visible trajectory for same period for an observer on the equator near
longitude 90°E (the point E in Fig.7).
Figure 7 shows the orientation of the Earth's crescent from the point of view of different
observers on the Moon and helps interpret the orientation of the libration pattern in Fig. 12.
The portion of the illuminated Earth is not changing with the position of the Moon observer, but
it's changing during lunar month (Fig.12, based on data by Espenak, 2021).
These arguments must be taken into account when planning observations of the Earth from the
Moon (or when communicating between the Moon and the Earth). Satellites located at the Earth-
Moon Lagrange points will move along similar trajectories in the sky of the Moon.
The principal design of the multislit spectrometer is shown in Fig. 13. Its main feature is that it
uses only one matrix detector for many slits. This is due to the fact that such a local object as the
Earth can pass only one slit at a given moment. Therefore, it is possible to image the light from all
slits onto a single matrix without compromising observations, although the problem of scattered
light may exist and should be studied in the development of a specific instrument. Each slit is
directed to a unique position of the Earth in the Moon sky, but the spectral dispersion of all slits is
the same. If the spectra are taken with a slit that occupies a length of 4000 pixels on the detector
matrix, then the spectra will be determined from the part of the Earth with a size of ~30 km along
the slit. The effective width of this pixel across the slit (i.e. spatial resolution) will depend on the
frequency of observations, the width of the slit, and the velocity of the Earth moving across the slit.

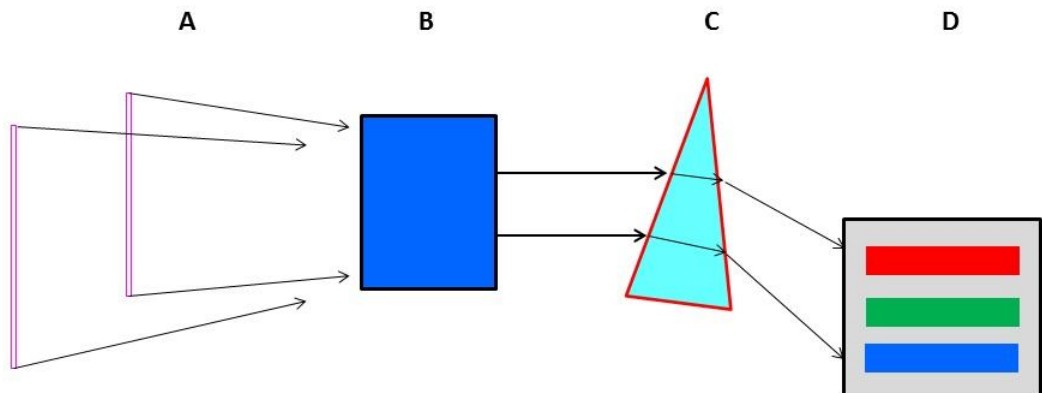

**Figure 13.** The principal design of a spectrometer that has multiple slits (A) and a single 2-
dimensional (2D) detector array (matrix). The spectrometer merges the light from all the slits
together, but since the Earth is always occupies only one slit, the signals from different slits do not
interfere with each other.
The angular diameter of the Earth in the sky of the Moon is about 1.9 degrees. The typical
rate of displacement of the center of the Earth is 1-2 degrees per day (see Figs. 8, 12). Therefore, the
passage of the Earth through each individual slit of the spectrometer will take 1-2 days. During this
time, the Earth makes 1-2 rotations around its axis, which will allow each slit to receive at least one
scan of the entire Earth's surface in one pass.
Important Earth science goals for such spectrometer are to complement and improve the
current DSCOVR/EPIC whole-Earth imaging (Gorkavyi et al., 2021). The acquired data will enable



estimating aerosol and cloud scattering phase functions, amount of trace gases and surface
Bidirectional Reflectance Factor (BRDF).

**5 Conclusion**
This paper discusses Earth observations from the Moon surface, both spectroscopically and in the
imaging mode. The librations of the Moon in the range of 13°-16° and the daily rotation of the Earth
serve as a natural scanning (or guiding) mechanism for a spectrometer with vertical slits. This
greatly simplifies the design of the spectrometer. We suggest that proposed lightweight EPIC-Moon
instrument on a fixed platform will provide the proof of concept for Earth observations, as well as
the whole Earth true-color imagery to the public.
The proximity to the Earth (versus the L1 point) and wide variations in phase angle accessible by a
Moon-based camera offer unique advantages for observations of the bidirectional land surface
reflectance; ocean/cloud glint reflection; whole-globe monitoring of transient volcanic/aerosol
clouds, polar mesospheric and stratospheric clouds; vegetation; the twilight zone and shadowed
parts of the Earth illuminated by the Moon.

**Data availability**
A Far-Ultraviolet Camera/Spectrograph data are available at https://gold.cs.ucf.edu/earths-shining-
upper-atmosphere-from-the-apollo-era-to-the-present/. The LRO data are available at
(https://www.nasa.gov/image-feature/goddard/lro-earthrise-2015 and at
(https://www.nasa.gov/feature/goddard/2020/moon-more-metallic-than-thought). The Apollo data
are available at https://moon.nasa.gov/news/38/nasa-mourns-the-passing-of-astronaut-john-young/
and at https://www.nasa.gov/mission_pages/apollo/missions/index.html. Planetary Ephemeris Data
Courtesy of Fred Espenak. Data are available at www.Astropixels.com.

**Author contributions.** NG developed computer codes and algorithms, analyzed the results, and
wrote the manuscript. NK, AM participated in algorithm development, analyzed the results, and
wrote the manuscript.

**Competing interests.** The contact author has declared that neither they nor their co-authors have
any competing interests.

**Acknowledgements.** The authors thank the Apollo, LRO and EPIC/DSCOVR teams for providing
the data presented. The authors are grateful to Fred Espenak for useful Planetary Ephemeris data.
We also thank Padi Boyd and Michele Gates from Goddard for their interest in the Earth
observations from the Moon surface.

**Financial support.** NK and AM were supported by the NASA DSCOVR project managed by
Richard Eckman. AM was supported by the Goddard Artemis project managed by Michele Gates.
NG was partially supported by the NASA Aura project (OMI core team) managed by Ken Jucks.

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
