# Peer review of "Earth Observations from the Moon surface: dependence on lunar libration"

_Atmospheric Measurement Techniques, 2022_

## Author Comment (AC1)

*Answers to RC1*

The manuscript entitled "Earth Observations from the Moon surface: dependence on lunar libration" provides innovative thoughts and visions on deploying DSCOVER/EPIC-Type camera on the moon for earth monitoring. Indeed, the Moon is a stable and longevous carrier for an Earth-observing sensor. It also exerts influences on precipitation, ice nuclei concentration, hurricanes, etc. For those Moon-related terrestrial phenomena, the lunar platform provides a unique perspective to understand the evolution of the phenomena.

Detailed analysis of the impact of the lunar libration on the visual position of the Earth on the Moon sky is presented, providing suggestions/insights on the possible field of view for the lunar-base earth observing sensors. This manuscript focuses on the impact of lunar libration on the sensor FOV. However, it would be more appealing if a more detailed discussion on the configuration of the potential lunar-base EPIC instruments.

Overall, I recommend the publication of this paper with minor revisions. The following are some minor points when I went through the paper in detail. They are only suggestions for the consideration of the authors when they revise the paper.

Line 9, line 293, and line 363, it will be better to have consistent values for the angular diameter of the Earth in the Moon sky.

*Answer 1: Done. The angular diameter of the Earth is 1.8°-2.0°*

Line 92, Libration of the Moon. I like the detailed explanation of optical libration provided here. It looks like the physical libration is neglected in the discussion due to its small magnitude compared to the optical libration.

*Answer 2: The following text has been added to the new version of the paper: "The libration of the Moon discussed above (Figures 3 and 4) is called optical libration. The tidal action of the Earth causes physical libration associated with a change in the period of the Moon's own rotation. Physical libration is only 2 arc minutes, that is, much less than optical libration. In the calculations (Figure 5 and below), physical libration is taken into account along with optical."*

Line 165, what does Figure serve for? I cannot find any discussion in the main text.

*Answer 3: Figure 6 has been removed.*

Line 285, what is the scientific goal for the slit observation? It will be better to provide some potential applications there to enlighten the reader.

*Answer 4: The following text has been added to the new version of the paper: "Spectrometric observations through the slit are a common practice for many satellite observations of the Earth. Scanning of the Earth's surface is usually carried out by movement of the low orbit satellites. For observation from the Moon, it is logical to consider the option when scanning occurs due to the libration and diurnal motion of the Earth. The scientific goals for the slit observation are close in both cases".*

Line 298, does this WFOV camera have the same spectral configuration as the EPIC?

*Answer 5: This will be determined in more detail later. The following insertion was made in the text of the paper: "The camera can be hyperspectral, with the inclusion of wavelengths that EPIC uses."*

Line 292, is 'wich' a typo?

*Answer 6: Fixed.*

Line 307, what does Figure 11 serve for? It looks like only the polar regions of the moon are plausible areas for the long-term operation of a lunar sensor. I suggest combining Figures 10 to 12 into one figure and focusing on one period to avoid unnecessary confusion.

*Answer 7: Figure 11 has been removed. The PRISM program consider the possibility of placing automatic instruments in the equatorial part of the Moon, which could work for several lunar days.*

Line 342, it will be better to provide the percentage of the loss of the lit area as the decrease of the Earth phase to give readers a better concept of the spatial and temporal observational capability of a moon-base sensor. I guess the Earth phase is one of the disadvantages of the lunar-base sensor. But if there is a dark/nighttime component in the instrument, it will greatly fill the gaps and provide valuable information on the dark side of the Earth just like VIIRS DNBs do.

*Answer 8: The information about the fraction of the illuminated areas has been added to Figure 12d. Observations of the night side of the Moon are possible, but this will depend on the specific instrument configuration, which has not yet been determined. The Earth phase is actually the advantage for the lunar-base sensor: a large range of phase angles is more important than the area of the Earth available for observation at any given moment. If the Earth is only partially illuminated, the diurnal rotation will cause the entire surface of the planet to pass through this illuminated portion as seen from the Moon.*

Line 363, What is the potential scan frequency of the WFOV camera and the hyperspectral sensor? Without this information, it is hard to follow the author's claim 'which will allow each slit to receive at least one scan of the entire Earth's surface in one pass. Please discuss more here if possible.

*Answer 9: The following text has been added to the new version of the paper: "The potential scan frequency depends on the field of view of the device and on the detector matrix used, so that the spatial pixel across the slit is comparable to the size of the spatial pixel along the slit. For a detector matrix with a size of 1000-4000 pixels and a field of view of 5 to 15 degrees, the scan frequency should be 10-100 seconds".*

Although it might be beyond the scope of this paper, it will be better to also shortly discuss/mention the possible limitation of the lunar-base sensor, such as the impact of the lunar environment (lunar dust exosphere, high energy cosmic particle, meteoroid, etc.) to give readers a whole picture of the concept.

*Answer 10: The following text has been added to the new version of the paper: "The lunar environment may have serious problems for the operation of sensors due to dust settling and impact on moving parts, and due to the influence of high-energy particles and meteoroids. The discussed design of instruments on a fixed mount with no movement of the external parts of the instruments, significantly reduces the dependence of observations on lunar dust. During the use of such instruments on the lunar surface, the rate of dust settling and the degree of degradation of*

*instruments due to radiation will be clarified, which will make it possible to optimize the design of future instruments and protect them as much as possible from a hostile environment".*

---

## Author Comment (AC2)

Referee comment on "Earth Observations from the Moon surface: dependence on lunar libration" by Nick Gorkavyi et al., Atmos. Meas. Tech. Discuss., https://doi.org/10.5194/amt-2022-158-RC2, 2022

General comments

This is a useful study, quantifying the Earth movement in the Moon sky, which can then be used to optimise Earth observations from the Moon. Based on this analysis, it emphasis the possibility of using a multi-slit based spectrograph, each slit associated with a different spectral range, and with a different position of the Earth in the sky.

*Answer 1: The new version of the article contains a clarifying comment: "Each slit is directed to a unique position of the Earth in the Moon sky, but the spectral range of all slits is the same."*

The authors should develop not only the advantage of a fixed observing platform on the Moon, but also the issues. For example, the libration is not always moving at the same speed, or following the same path (i.e. sometimes the Earth may not go through some of the slits; sometimes, it may stay on one slit, then return).

*Answer 2: Most of the article is devoted to a discussion of the complex trajectory of the Earth, with accelerations, stops and returns. In the new version of the paper (part 4), the difficulties of comparing data from different slits, even if they work in the same spectral range, are discussed: "The angular velocity of a point on the Earth's surface in the field of view of the sensor is caused by two comparable factors: the rotation of the Earth around its axis and lunar libration, which causes a shift in the center of the Earth. The rotation of the Earth around its axis is a well-studied process, but librations of the center of the Earth in the lunar sky are poorly understood and raise many questions. When observing the Earth through the slit of the spectrometer, it will be necessary to take into account both the displacement of the center of the Earth and the Earth rotation."*

Scanning different spectral ranges at different times may not be relevant to compare them.

*Answer 3. All slits have the same spectral ranges (see Answer 1).*

The quality of the graphs in the draft is poor, raster at around 140 dpi. For the final paper please use vector based graphics or raster of at least 300 dpi!

*Answer 4. The quality of the figures in the new version has been substantially improved.*

Specific comments

31 – Forward direction is around $0^{o}$, while backscatter direction is by definition around $180^{o}$.

*Answer 5. In astronomy another definition of phase angle is used and the forward scattering is close to 180 deg – see "Phase_angle_(astronomy)" (Wiki)*

Then, the phase angle cannot be between $2^{o}$ and $12^{o}$. Also, why is starting at $2^{o}$ and not at $0^{o}$? In addition, why is that much anyway if the size of the Earth as seen from L1 is $< 1^{o}$. Even if these affirmations come from other papers, they seem so strange that the reader may question if they are true. Please be proactive and give more info (for example, by answering my questions above) to convince the reader.

*Answer 6. An explanation has been added to the new version of the paper: "The phase angle interval from 2°
to 12° is determined by the trajectory of the DSCOVR space observatory, which does not rest at the Lagrange
point, but moves around it". It is too 'noisy' to transmit data directly from the Earth-Sun line; thus, the phase
angle is bigger than 2°.*

81 – Still concerning the Figure 2, I know you had this one for a long time in the manuscript, but I realise
now that it is misleading to have the Epic image as an "insert" of smaller dimensions. It suggests you may
have such a large change in angular dimension of the Earth between LRO and Epic. I suppose the Earth
angular size should be similar in both. Then I would rather suggest to put those pictures side by side in Fig 2,
with the same size of Earth.

*Answer 7. An explanation has been added:   "The difference in distances Earth - Moon and Earth – L1 leads
to the fact that a telescope with the same field of view sees the Earth with different angular sizes and
resolutions".*

167 - Figure 6 was never introduced or discussed in the text. The figure should be introduced after its first
mention.

*Answer 8. Figure 6 has been removed.*

243 – Here you give a definition of the "draconic month" different from that of line 100. They may be
equivalent, but that is not necessarily obvious for the reader. Please use the same definition, or explain why
the two definitions are equivalent.

*Answer 9. Thanks for the helpful note. The definitions of the draconian and sidereal months were mistakenly
swapped. They have now been corrected.*

374 – I think you could remind in the conclusions more from your results that could be useful messages for
the reader, like the statistics (density) of points, speed of libration etc. Otherwise, it looks as you didn't find
much.

*Answer 10. The conclusion section has been revised. In particular, the following items have been added:*

*"1. Due to lunar libration, the center of the Earth for an observer on the Moon moves in a rectangular area
with dimensions of 13.4° × 15.8°. The density of the location of the Earth in this rectangular area is an
average of 8.6 per square degree over 6 years (2191 days). The density for different parts of the area varies
from 1 to 20.*

*2. The movement of the Earth in the sky of the Moon is characterized by quasi-periodicity with frequencies of
~27 days and 6 years. The rates of displacement of the Earth in the Moon sky reach two degrees per day. The
shape of the Earth's trajectory changes from a circle to a straight line (see Figure 7).*

*3. Lunar libration must be taken into account when observing the Earth from the surface of the Moon and
during Moon-Earth communications".*

Technical comments

1 – This comment is just an advice to avoid using ":" in the paper title "Earth Observations from the Moon surface: dependence 1 on lunar libration". This is a special formatting character in Latex and it usually leads to errors when citing and referencing your paper.

*Answer 11: Thank you*

9 – "1.9$^o$ +/- 0.1$^o$ " has to be in italic as well, if the entire Abstract is.

*Answer 12: Fixed.*

12 – Add: reaching "an amplitude of" 7.9$^o$ .

*Answer 13: Added.*

13 – Add: center of the Earth "to" move

*Answer 14: Added.*

53 – The resolution of Fig 1 is 142 pixels/inch and the journal requires minimum 300 pixels/inch. If those pictures are not available at higher resolution, you may increase by resample it artificially to respect the rule.

*Answer 15. The quality of the figures has been improved.*

81 – Same as for 53. I suspect your pdf maker was set to a lower resolution. Please check and, if true, increase its resolution rather than resampling the pictures.

*Answer 16: See answer 15.*

84-85 – I think you mean "a linear (one-dimensional) array of 5064 elements." instead of

"the linear (one-dimensional) array from 5064 elements."

*Answer 17: Fixed.*

114 – Concerning "between 0.0255÷0.0775", I do not think the sign "÷" is correctly used. It is more for calling a "division". You should better use "between 0.0255 and 0.0775".

*Answer 18: Fixed.*

167 - Figure 6 is also of a too low resolution.

*Answer 19. Figure 6 has been removed.*

183 – Why do you have a break of line (row) here?

*Answer 20. See answer 19.*

191 – Replace "over a given point" with "above a given point".

*Answer 21: Fixed.*

266 – You forgot the verb "is" N = 2191.

*Answer 22:  Fixed.*

283 – You have to be consistent when calling the Figures 10-11 (as everywhere else), not Figs 10-11.

*Answer 23:  Done, thank you.*

336-338 – Use Figure 7 not Fig. 7.

*Answer 24:  Done.*

340-343 – Use Figure 12 not Fig. 12.

*Answer 25:  Done.*

378-379 – "We suggest that proposed lightweight EPIC-Moon instrument on a fixed platform will provide the proof of concept" is ill formulated. May be you want to say "We suggest a lightweight EPIC-Moon instrument on a fixed platform to serve as a proof of concept", or something similar.

*Answer 26: Done, thank you.*

---

## Referee Report (RR1)

Please send the revised manuscript with all the changes emphasised!
* * *
- REVIEW 1

Concerning my comment at line 31:

31 – Forward direction is around 0°, while backscatter direction is by definition around 180o.

with the he authors *Answer 5*: "In astronomy another definition of phase angle is used and the forward scattering is close to 180 deg – see "Phase_angle_(astronomy)" (Wiki)".

REVIEW:

I do not agree with this explanation, as this is not an astronomy paper, but an Earth remote sensing one. In this context people expect to preserve their conventions, otherwise it is confusing. Even your own reference to DSCOVR EPIC (https://doi.org/10.3389/frsen.2021.719610) explicitly mentions

"reaching 178°. This provides a unique opportunity to observe bi-directional effects of reflectance near backscattering directions".

So 180 deg is considered backscatter in your own reference. I propose to stay consistent and use remote sensing convention, to make sure there is no confusion in the remote sensing community. You may eventually also provide the definition of the phase angle convention in a footnote, if you want to avoid confusion to a larger audience.
* * *
- REVIEW 2

Concerning the authors *Answer 10. The conclusion section has been revised. In particular, the following items have been added:*

"1. Due to lunar libration, the center of the Earth for an observer on the Moon moves in a rectangular area with dimensions of 13.4° × 15.8°. The density of the location of the Earth in this rectangular area is an average of 8.6 per square degree over 6 years (2191 days). The density for different parts of the area varies from 1 to 20."

REVIEW:

Please specify what the density units are, otherwise it is meaningless. In order to help this out, you also need to do the following:

Figure 9a - remind that each point is the averaged Earth location over one day (or location at noon, or whatever you actually did).

Figure 9b - provide units for the density of points (i.e. days spent in that pixel of 1x1 deg solid angle). Also <N>=8.6 should be with the average sign, while the actual N is from 0 to 34. They are not the same thing.

---

## Author Response (AR2)

January 20, 2023

**- REVIEW 1**

Concerning my comment at line 31:

31 – Forward direction is around 0 o , while backscatter direction is by definition around 180o.

with the he authors Answer 5: "In astronomy another definition of phase angle is used and the forward scattering is close to 180 deg – see "Phase_angle_(astronomy)" (Wiki)".

REVIEW:

I do not agree with this explanation, as this is not an astronomy paper, but an Earth remote sensing one. In this context people expect to preserve their conventions, otherwise it is confusing. Even your own reference to DSCOVR EPIC (https://doi.org/10.3389/frsen.2021.719610) explicitly mentions "reaching 178°. This provides a unique opportunity to observe bi-directional effects of reflectance near backscattering directions".

So 180 deg is considered backscatter in your own reference. I propose to stay consistent and use remote sensing convention, to make sure there is no confusion in the remote sensing community. You may eventually also provide the definition of the phase angle convention in a footnote, if you want to avoid confusion to a larger audience.

ANSWER 1.

Apparently, the reviewer is mixing up "scattering angle" and "phase angle". In the mentioned article (https://doi.org/10.3389/frsen.2021.719610) (its first author is a co-author of our paper and the deputy project scientist for the DSCOVR mission), the phrase "reaching 178°" refers to "scattering angle", which is associated with "phase angle" following relation (quote from mentioned article):

"We will be using "scattering angle" here which goes from $0°$ (forward scattering) to $180°$ (backward scattering). We will be also using Sun Earth Vehicle (SEV) angle (a.k.a. Phase angle) = $180°$–scattering angle."

To avoid confusion, on the first page of the new version of the manuscript, we made an addition:

"The L1 location, however, limits phase (i.e., Sun-Earth-Camera) angles to a nearly backscattering direction (up to ~$178°$)."

**- REVIEW 2**

Concerning the authors Answer 10. The conclusion section has been revised. In particular, the following items have been added:

"1. Due to lunar libration, the center of the Earth for an observer on the Moon moves in a rectangular area with dimensions of 13.4° × 15.8°. The density of the location of the Earth in this rectangular area is an average of 8.6 per square degree over 6 years (2191 days). The density for different parts of the area varies from 1 to 20."

REVIEW:

Please specify what the density units are, otherwise it is meaningless. In order to help this out, you also need to do the following:

Figure 9a - remind that each point is the averaged Earth location over one day (or location at noon, or whatever you actually did).

Figure 9b - provide units for the density of points (i.e. days spent in that pixel of 1x1 deg solid angle). Also <N>=8.6 should be with the average sign, while the actual N is from 0 to 34. They are not the same thing.

ANSWER 2 about Fig.9a:

This figure (now numbered 8a) shows the line (without points) along which the Earth moves for 6 years.

ANSWER 3 about Fig.9b:

In the caption to this figure (now 8b), we marked the mean value of $N$ with the traditional sign for the arithmetic mean: $\overline{N}$. Added to the figure caption:

"(or how many days the Earth spent in a 1°×1° pixel)".